# Personalizing the empiric treatment of gonorrhea using machine learning models

**Rachel E. Murray-Watson**[1,2]*, **Yonatan H. Grad**[3], **Sancta B. St. Cyr**[4], **Reza Yaesoubi**[1,2]

**1** Department of Health Policy and Management, Yale School of Public Health, New Haven, Connecticut, United States of America, **2** Public Health Modeling Unit, Yale School of Public Health, New Haven, Connecticut, United States of America, **3** Department of Immunology and Infectious Diseases, Harvard T. H. Chan School of Public Health, Boston, Massachusetts, United States of America, **4** Division of STD Prevention, Centers for Disease Control and Prevention, 1600 Clifton Road NE, Atlanta, Georgia, United States of America

* rachel.murray-watson@yale.edu

## Abstract

Despite the emergence of antimicrobial-resistant (AMR) strains of *Neisseria gonorrhoeae*, the treatment of gonorrhea remains empiric and according to standardized guidelines, which are informed by the national prevalence of resistant strains. Yet, the prevalence of AMR varies substantially across geographic and demographic groups. We investigated whether data from the national surveillance system of AMR gonorrhea in the US could be used to personalize the empiric treatment of gonorrhea. We used data from the Gonococcal Isolate Surveillance Project collected between 2000–2010 to train and validate machine learning models to identify resistance to ciprofloxacin (CIP), one of the recommended first-line antibiotics until 2007. We used these models to personalize empiric treatments based on sexual behavior and geographic location and compared their performance with standardized guidelines, which recommended treatment with CIP, ceftriaxone (CRO), or cefixime (CFX) between 2005–2006, and either CRO or CFX between 2007–2010. Compared with standardized guidelines, the personalized treatments could have replaced 33% of CRO and CFX use with CIP while ensuring that 98% of patients were prescribed effective treatment during 2005–2010. The models maintained their performance over time and across geographic regions. Predictive models trained on data from national surveillance systems of AMR gonorrhea could be used to personalize the empiric treatment of gonorrhea based on patients' basic characteristics at the point of care. This approach could reduce the unnecessary use of newer antibiotics while maintaining the effectiveness of first-line therapy.

**Data Availability Statement:** The data underlying the results presented in the study are available

## Author summary

Treating gonorrhea is complicated by the spread of drug-resistant strains, yet current approaches rely on empiric guidelines. Our study explored using national data on resistance in gonorrhea to personalize treatment decisions. From 2000 to 2010, we analyzed information from the Gonococcal Isolate Surveillance Project to develop predictive models. These models considered factors like where patients live and their behavior to suggest

from the Gonococcal Isolate Surveillance Project (https://www.cdc.gov/std/gisp/default.htm).

**Funding:** This work was supported by the National Institute Of Allergy And Infectious Diseases of the National Institutes of Health (R01AI153351 to RY). The funders had no role in study design, data collection and analysis, decision to publish, or preparation of the manuscript.

**Competing interests:** The authors have declared that no competing interests exist.

treatments tailored to their likely resistance profile. Compared to standard guidelines during that period, our personalized approach could have reduced unnecessary use of certain antibiotics by 33%, while still effectively treating 98% of patients. These findings highlight the potential of using data-driven models to improve how we treat gonorrhea, ensuring effective care while safeguarding newer antibiotics for the future.

## Introduction

Antimicrobial-resistant (AMR) gonorrhea is a pressing issue. *Neisseria gonorrhoeae*, the causative agent of gonorrhea, has proven adept at evolving or acquiring resistance mechanisms to all antibiotics that have been used to treat it [1]. In 2019 alone, there were an estimated 942,000 cases of AMR gonococcal infections [2], an increase of 71.3% on the 2017 estimate. This makes it one of the most urgent antibiotic-resistant threats in the United States (US) [3].

Though the susceptibility of gonococcal strains to different antibiotics can be determined using bacterial cultures, they have mostly been replaced by nucleic acid amplification tests (NAATs) in the diagnosis of gonorrhea [4]. Without antimicrobial-specific diagnostic information, gonorrhea is treated empirically according to standardized treatment guidelines, which are determined at the national level and based on estimates for the prevalence of AMR gonorrhea. In the US, for example, these estimates are provided by the Gonococcal Isolate Surveillance Project (GISP), a national surveillance system that collects data from an average of 28 sentinel sites in the US [5].

To ensure the effectiveness of empiric first-line therapy, guidelines recommend antibiotics with a low national prevalence of resistance (usually <5%) for all patients with gonorrhea [6–8]. However, reports of substantial variation in the AMR prevalence and trend among different geographic regions or population subgroups [9–13]. Given the empiric nature of treatment, these variations led to large disparities in the number of patients receiving effective treatment. Such concerns resulted in revised guidelines that accounted for these variations. For example, in 2000, estimates from GISP indicated that there was an increase in the prevalence of ciprofloxacin-resistant (CIP-R) gonococcal infections in Hawai'i. In response, CDC revised the guidelines for the state of Hawai'i, no longer recommending CIP as the first-line treatment[6]. Two years later, in 2002, CIP was also removed from the treatment recommendations for gonorrhea in California. By 2004, the advice against the empirical use of CIP was extended to MSM, and finally, in the year 2007, the CDC recommended against using CIP for all patients with gonorrhea.

These refinements highlight the need for a principled approach to personalize empiric treatment based on patient characteristics that are associated with the risk of AMR gonorrhea (such as sexual behavior and geographic location).

In this study, we examined whether data from GISP could be used to develop predictive models that enable personalized empiric treatment of gonorrhea based on a select set of patient characteristics, including sexual behavior and geographic location, in addition to information related to the local prevalence of AMR gonorrhea. We estimated the effectiveness of these personalized treatment recommendations and their ability to reduce the unnecessary use of newer antibiotics where older treatments were effective, compared with standardized treatment guidelines.

## Methods

Our dataset includes the record of isolates evaluated for resistance as part of GISP between 2000–2010. Every month during this period, GISP collected isolates from the first 25 men

diagnosed with urethral gonorrhea from 33 sentinel sites across five geographic regions in the US (Table A in S1 File) [14]. Information on sexual behavior (determined by the sex of the sexual partner) was collected from each individual. These isolates are then tested for susceptibility to ciprofloxacin (CIP), cefixime (CFX), ceftriaxone (CRO), and other antibiotics. In this study, we use the GISP data between 2000–2010 to develop and evaluate models to predict reduced susceptibility to CIP for an individual with gonorrhea. Reduced susceptibility for CIP is defined as presenting either resistance to CIP (a minimum inhibitory concentration of $\geq 1\mu g/mL$) or intermediate resistance to CIP (a minimum inhibitory concentration of $0.125–0.5\mu g/mL$) [14]. Our analysis encompasses the period when CIP was one of the recommendations for first-line therapy for gonorrhea, alongside CRO and CFX. CIP was first recommended for the treatment of gonorrhea in 1993, but by 2006, about 13% of gonorrhea isolates evaluated by GISP were resistant to CIP [6]. In April 2007, the CDC stopped recommending CIP for the empiric treatment of gonorrhea. Instead, between 2007 and 2010, either ceftriaxone or cefixime ("CRO/CFX") was recommended as first-line treatment [6].

Our objective is to examine, retrospectively, the effectiveness of treatment recommendations that are personalized based on the patient's geographic location and sexual behavior, and information about the local and regional epidemiology of AMR gonorrhea. We also compare the effectiveness of these personalized treatment recommendations and their ability to reduce the unnecessary use of CRO and CFX with treatment guidelines recommended by CDC during this period.

## Predictors and outcomes

To develop our predictive models, we considered eight features: 1) the sexual behavior of the individual who submitted a sample, 2) the geographic region in which the sample was taken, 3) the prevalence of resistance to CIP among the isolates collected last year by the surveillance site, 4) the prevalence of resistance to CIP among the isolates collected last year from the geographic region where the surveillance site is located, and 5 and 6) the change in the prevalence of resistance from the previous year in the surveillance site and the geographic region. Given the variation in the annual number of samples collected from each surveillance site, we included two additional binary features to account for the accuracy of estimates for the prevalence of resistance to CIP: 7) whether the prevalence data was calculated on more than 75 observations for that year, and 8) whether the trend data was calculated on more than 75 observations for each year. These features were based on data made available as part of the GISP survey. Other variables of interest, such as age, or ethnicity, were not available as part of this dataset.

Two variables (region and sexual orientation) were nominal categorical variables, and we used one-hot encoding to include them in our analysis (§§S1.1 of SI). The geographic location was available for all individuals. For sexual orientation, we coded entries with missing, other, or unknown as "missing" and kept them in the analysis. We standardized all continuous variables (the previous year's regional and clinical prevalence of CIP resistance and the change in regional and clinical prevalence).

The outcome we wanted to predict was reduced susceptibility to CIP for a gonococcal infection given the patient's characteristics (e.g., geographic location and sexual orientation). Therefore, all samples with CIP resistances, including samples with additional resistance to other antibiotics, were labelled as positive instances. During our study period (2000–2010), the prevalence of resistance to each of CRO and CFX was <0.01%. Hence, we assume that all gonococcal infections during this period were susceptible to CRO and CFX and could be successfully treated with these antibiotics [5].

## Model development

As described above, the guideline for the empiric treatment of all individuals with gonorrhea was changed in 2007 to remove CIP from the recommended treatment guidelines, leaving only CRO or CFX available as recommended treatments. Our goal is to evaluate the utility of personalized treatment recommendations during years before and after the change in recommendation occurred. As such, for each year between 2005–2010, we used data during the prior five years to develop and validate predictive models to inform personalized treatment recommendations during the year of interest, which we refer to as the "target year". We restricted the training set to just five years as there were large changes in the prevalence of CIP resistance, so using data from more than five years before the target year in question may have reduced the accuracy of our models.

To develop predictive models for each target year, we followed the procedures outlined in the Transparent Reporting of a multivariable prediction model for Individual Prognosis Or Diagnosis (TRIPOD) [15]. For each year, we considered three supervised Machine Learning (ML) models: a multilayer perceptron (neural network), a logistic regression model, and a random forest. We developed all of our machine learning models using the scikit-learn package in Python [16].

The predictive models we considered here estimate the probability that a gonococcal infection has reduced susceptibility to CIP, given patient characteristics. If this probability is greater than a classification threshold $p$, we classified the infection as "CIP-non-susceptible"; otherwise, the infection is classified as "CIP-susceptible". Changing the classification threshold, $p$, allows us to change the sensitivity and specificity of each model and quantify the trade-off between the model's ability to recommend effective personalized treatment versus to prevent the unnecessary prescription of CRO/CFX when the infection was susceptible to CIP.

To improve the predictive power of each year's models, we performed hyperparameter tuning using a random search algorithm and ten-fold cross-validation, which has been shown to reduce the bias and variance of predictive models [17] (§S1.2 of SI). We then used permutation importance to determine which features increased the area under the Receiver Operating Characteristic curve (auROC). After removing the features that decreased the auROC, we conducted another round of hyperparameter tuning using 10-fold cross-validation for each model in each target year.

## Model validation and performance

In line with TRIPOD recommendations [15], we used temporal validation to assess the performance of our models. For each target year, data from the five previous years was used for training the model, hyperparameter tuning, and feature selection. We then used the records collected during the target year to evaluate the model performance based on three criteria: the area under the receiver operating characteristic curve (auROC), the Matthews correlation coefficient (MCC), and the F1 score. As the values of the latter two depend on the classification threshold ($p$), we calculated scores over a range of classification thresholds.

To evaluate the sensitivity of our estimates to the observations collected during training years, the selected features, and the selected values for each model's hyperparameters, we also report 95% bootstrap confidence intervals, calculated using the method described in §S1.3.

We also conducted Leave-One-Out Cross Validation (LOOCV) to evaluate model generalizability beyond surveillance sites included in GISP. For each target year, we first developed models using the data from all surveillance sites excluding data from one of the sites. Then, we temporally validated the models using observations from the excluded site submitted during the target year.

### Effectiveness of personalized treatment recommendations and their impact on the unnecessary use of CRO/CFX

For each target year, we report the percentage of individuals that would 1) receive effective treatment (i.e., the treatment that matches the resistance profile of their gonorrhea strain) and 2) be unnecessarily treated with CRO/CFX therapy while their infecting strain is susceptible to CIP. For example, following the standardized guidelines between 2007–2010, which recommended treatment with CRO/CFX, would result in 100% treatment effectiveness but at the cost of unnecessarily using CRO/CFX for $100(1 - \mu)$% for individuals with gonorrhea, where $\mu$ is the prevalence of resistance to CIP in that year. The additional details on how these measures are calculated are provided in §S1.5 of SI.

These metrics allow us to estimate the change in the treatment effectiveness and the unnecessary use of CRO/CFX if personalized treatments are followed instead of standardized guidelines. We also included a comparison with the count of individuals who would have received effective treatment under the revised CDC guidelines years 2005–2006, which recommended against using CIP for those residing in California and Hawai'i and those identifying as MSM.

## Results

Among the isolates collected during 2000–2010, 8.1% had resistance to CIP. The prevalence of resistance to CIP was highest in the West and among the MSM population (Table 1). Models developed for each target year had an auROC greater than 0.5, indicating that they were better than random chance at predicting reduced susceptibility to CIP (Fig 1). The higher the auROC values, the better the model was at distinguishing between ciprofloxacin-susceptible and resistant strains. Models developed for each target year had an auROC greater than 0·5, indicating that they were better than random chance at predicting reduced susceptibility to CIP (Fig 1). The higher the auROC values, the better the model was at distinguishing between ciprofloxacin-susceptible and resistant cases. The neural network and random forest models had comparable performance based on the Matthews correlation coefficients (MCCs) and F1 scores but outperformed the logistic regression models (Fig B in S1 File). Based on these metrics, the neural network model had the highest and most consistent performance scores across target years, indicating that it could respond to changes in the prevalence of CIP-R across time. We, therefore, focus the remainder of our analyses on neural network models, though results for the other ML models are shown in §S2 of SI.

For the neural network models, the most important features included whether the patient was MSW or MSM, the local prevalence of resistance to CIP ("Local CIP-R"), and the change

**Table 1. Select demographic information of GISP isolates collected between 2000–2010 ($N$ = 64,445).** Overall, for this period, 8.08% of isolates had resistance to CIP.

| Variable | % of isolates with resistance to CIP | % of all isolates |
|---|---|---|
| Geographic region | | |
| Midwest | 1·6 | 21·2 |
| Northeast | 7·7 | 10·9 |
| Southeast | 5·1 | 18·5 |
| Southwest | 4·6 | 13·9 |
| West Sexual behavior | 14·9 | 35·5 |
| MSM | 23·6 | 17·0 |
| MSW | 4·3 | 76·1 |
| MSMW | 17·7 | 3·6 |
| Unknown/Missing | 4·5 | 3·4 |

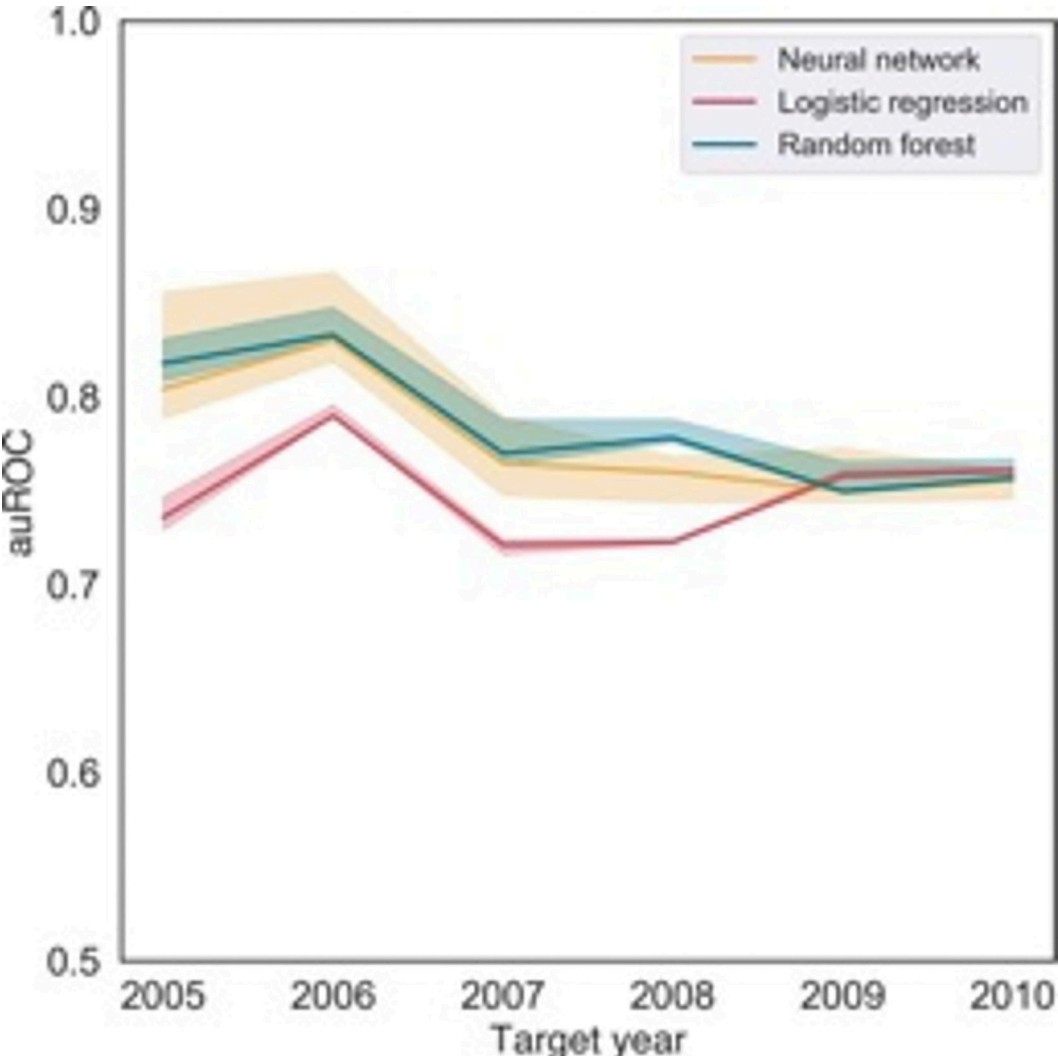

**Fig 1. Area under the receiver operating curve (auROC) for each predictive model.** For each target year, the model was trained on data from the five previous years and then applied to the data collected during the target year to calculate auROC. The shaded regions show the 95% bootstrap confidence interval, calculated as per the algorithm in S1 File.

in the local and regional CIP-R prevalence ("Local/Regional change in CIP-R") (Fig 2). Other highly important features included the region in which the sample was collected. For each target year, only features that either improved the auROC or did not affect it were included in the final model. Included features and their importance for the logistic regression and random forest models are shown in Fig B in S1 File.

Under standardized guidelines, the proportion of patients receiving effective treatment or being unnecessarily treated with CRO/CFX depended on the proportion of patients who were prescribed CIP versus CRO/CFX. This is represented by the grey dashed lines in Fig 3; in each panel, the scenario where all patients are treated with CIP (which we refer to as "CIP") is represented by a star, and the scenario where all patients are treated with CRO or CFX (which we refer to as "CRO/CFX") is represented by a black box. These lines represent the performance of standardized guidelines when a portion of individuals receive CIP, and the remaining portion receives CRP/CFX.

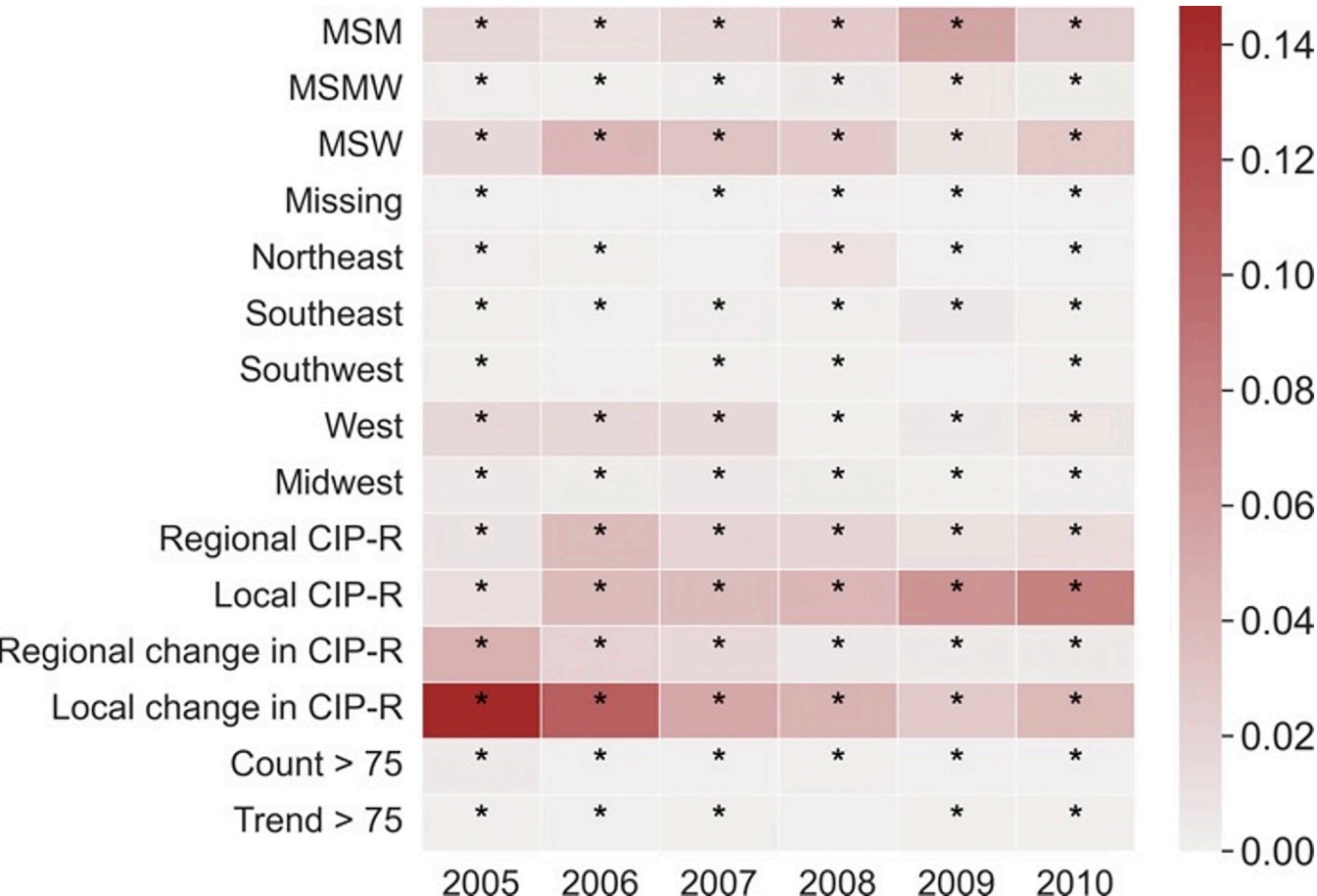

**Fig 2. The contribution of features to the area under the receiver operating curve (auROC) score across each model and target year.** The color bar represents the contribution of each factor, measured by the feature importance score provided by the permutation importance algorithm. Only features with a positive effect on the training auROC were used in the final models. These features are represented by '*'. CIP-R is the prevalence of resistance to ciprofloxacin.

The "X" marks the CDC's revised treatment guidelines where the majority of patients were prescribed CIP but patients in Hawai'i, California, and those identify as MSM received CRO or CFX. Compared to scenarios "CIP" or "CRO/CFX", these revised CDC guidelines improved the percentage of individuals receiving effective treatment but also increased the unnecessary use of CRO/CFX.

For each target year, for a given threshold for treatment effectiveness (say, 98%, which is represented by vertical black dotted lines in Fig 3), following the personalized treatment provided by neural network models could have reduced the unnecessary use of CRO/CFX. The magnitude of this reduction is measured by the vertical distance between the dotted line and the bold curve in each panel of Fig 3 and is dependent on the model's classi- fication threshold ($p$), represented by numbers on curves in Fig 3. For each target year during 2005–2010, we could identify a classification threshold such that the effectiveness of personalized treatment regimens is maintained at 98%. At this effectiveness threshold, fol- lowing the personalized recommendations provided by the neural network model could have reduced the unnecessary use of CRO/CFX by an average of 32.8 percentage points compared to recommendations pro- vided by standardized guidelines during 2005–2010.

The results of Leave-One-Out Cross Validation (LOOCV) suggest that the ability of pre- dictive models to reduce the unnecessary use of CRO/CFX while maintaining a high level of

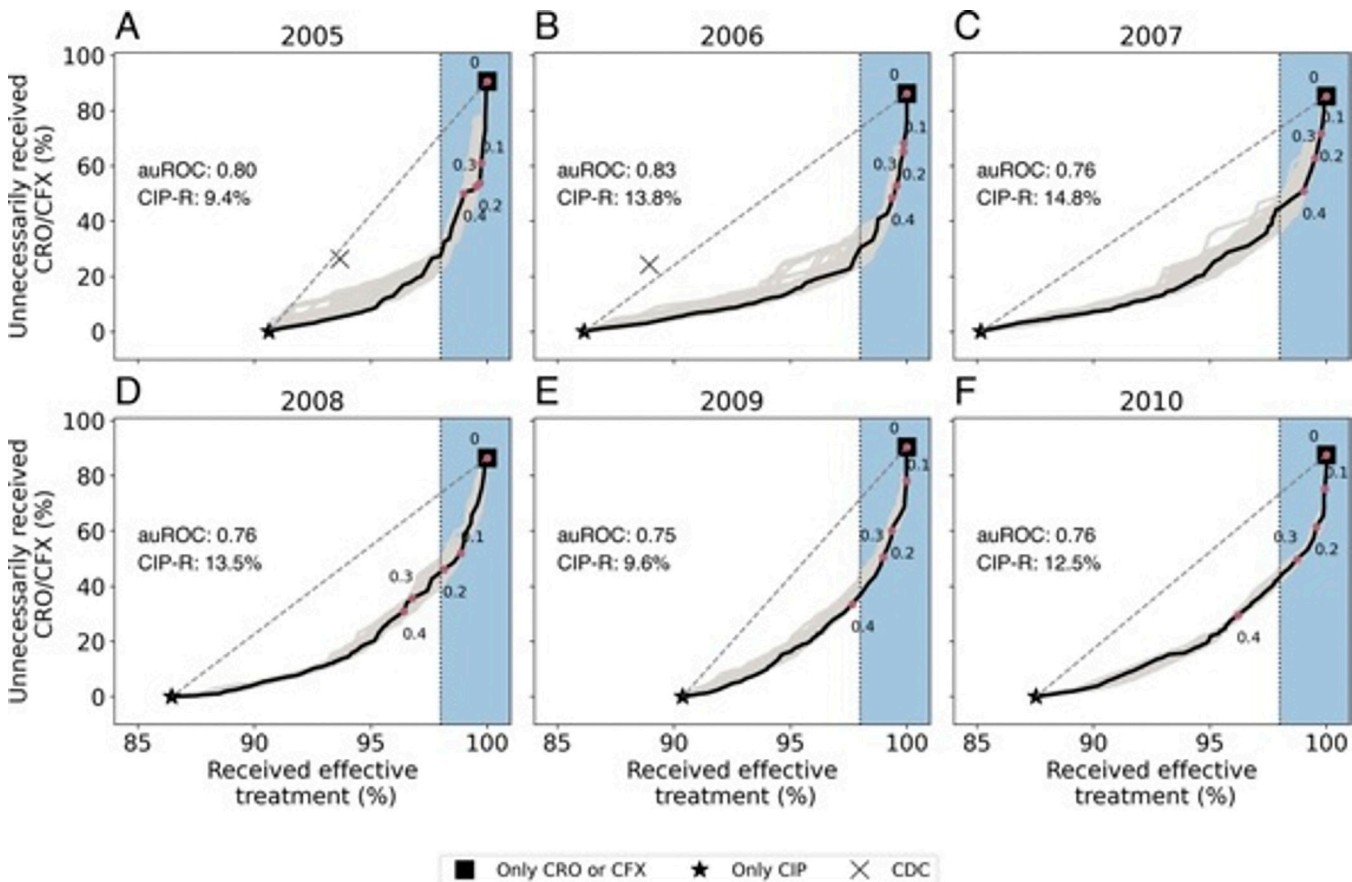

**Fig 3. The proportion of patients with gonorrhea receiving effective treat- ment or being unnecessarily treated with CRO/CFX when CIP could have been effective.** Numbers on the curves represent different classification thresholds. Higher clas- sification thresholds decrease the proportion receiving effective treatment and unnecessarily treated with ceftriaxone or cefixime (CRO/CFX). The grey dashed line shows the impact if a random portion of individuals with gonorrhea receive CIP and the rest receive CRO/CFX therapy. The "X" denotes the scenario where no one in California, Hawai'i, or who identified as men who had sex with men (MSM) received ciprofloxacin (CIP) as recommended by CDC treatment guidelines in 2005 and 2006. The blue regions represent predictive models that would result in 98% treatment effectiveness. The grey curves show the 95% bootstrap region generated using the bootstrapping procedure described in S1.3.

treatment effectiveness varies across different locations (Fig 4). At low classification thresholds, the proportion receiving effective treatment can be maintained at > 95% while reducing the unnecessary prescription of CRO/CFX by between 4 and 100 percentage points, in some cases, therefore, eliminating the unnecessary prescription of CRO/CFX. However, for regions with higher than the national average CIP-R prevalence (i.e., pink curves in Fig 4), the models trained on national data would have been less effective in reducing the unnecessary use of CRO/CFX while maintaining the treatment effectiveness at 95%.

## Discussion

The common methods to diagnose gonorrhea do not provide information about the suscepti- bility of the infecting strain to available antibiotics [4,18]. The treatment of gonorrhea, there- fore, remains empiric and according to standardized guidelines, which are determined at the national level. Although these standardized guidelines are easy to communicate, they may result in prescribing an ineffective therapy and diminishing the effective lifespan of existing antibiotics. In this retrospective study, we used data from GISP collected between 2000–2010

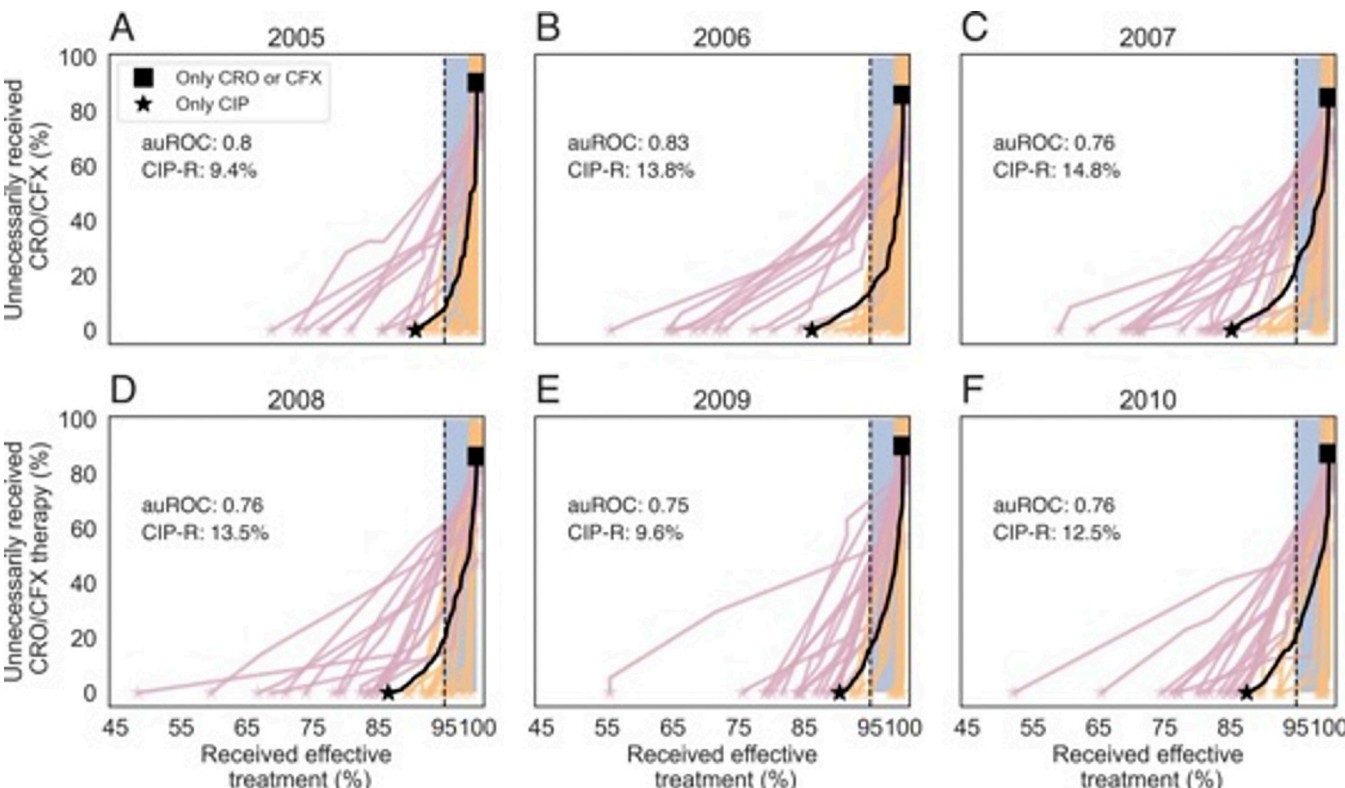

**Fig 4. Leave-One-Out Cross Validation (LOOCV).** Each individual curve corre- sponds to the surveillance site the model was tested on, but the data from this surveillance site was excluded when developing the model. Pink lines are clinics where the resistance to ciprofloxacin (CIP-R) prevalence was above the national average, and orange lines are where it was below. The black curve corresponds to the model that is developed using data from all surveillance sites during the five years prior to the target year. The blue regions represent predictive models that would result in 95% treatment effectiveness.

to develop and evaluate machine learning (ML) models that use basic demographic informa- tion to identify reduced susceptibility to CIP, one of the recommended first-line antibiotics in the US until 2007 [6]. We estimated the impact of following the recommendations provided by these predictive models on the probability of using effective antibiotics and the unnecessary use of either ceftriaxone or cefixime (CRO/CFX), which were the recommended first-line ther- apies between 2007–2010 [6].

Our analysis showed that following the personalized recommendation provided by these ML models could have improved the probability of receiving effective treatment and reduced the unnecessary use of CRO/CFX among patients with gonorrhea between 2005–2010 in the US. In the neural network and random forest models, we found that regional and local epide- miological factors were generally the most important feature, alongside sexual behavior (Fig 2 and Fig B in S1 File). These features are consistent with previous studies suggesting that CIP resistance is more common amongst some "core" groups defined by sexual behavior [14,19,20], and that the local prevalence of resistance is an important factor in determining empiric treatment guidelines [14,21]. However, in both models, the majority of features were considered important (having a positive impact on the auROC). Considerably fewer features were included in the logistic regression model, though important features included sexual behavior and the region in which the sample was collected.

Of the three models developed (logistic regression, neural network, and random forest), the neural network model had the highest and most stable auROC (Fig 1), MCC and F1 scores across all test years (SS2 in SI). We, therefore, focus the majority of our analyses on the results

of this model. The ability of the models considered here to improve treatment effectiveness and reduce the unnecessary use of newer antibiotics depended on the selected classification threshold. Across all classification thresholds, there was a trade-off between the proportion of patients receiving effective treatment and the unnecessary use of CRO/CFX (Fig 3 and Figs F-G in S1 File). Yet all models could have led to significant reductions in the unnecessary use of CRO/CFX while maintaining the efficacy of first-line treatment above 98%.

Between 2000 and 2007, the CDC revised its empiric treatment guidelines in response to an increasing prevalence of CIP-resistant infections. Whilst these revised guidelines improved the proportion of patients receiving effective treatment, they also increased the proportion that were unnecessarily treated with CRO/CFX when CIP would have been effective. The approach that we proposed here provides a more systematic method to translate data collected through GISP into personalized treatment recommendations, reducing the unnecessary use of new antibiotics without affecting the proportions of receiving effective treatment.

The optimal choice of classification threshold requires quantifying the positive and negative consequences of using predictive models with imperfect sensitivity and specificity (Figs C-E in S1 File). The acceptable reduction in treatment efficacy for a given reduction in the unnecessary use of CRO/CFX may vary between decision-makers. This could be more carefully investigated through a cost-effectiveness analysis.

Our study has several limitations. First, our dataset is limited only to the surveillance sites included in GISP, and hence the performance of predictive models developed here for locations not included in GISP remains unknown. However, using LOOCV, we found that when the model was used to make predictions for clinics not included in the training set, treatment efficacy could remain > 95% while reducing–and in some cases eliminating–the need for CRO/CFX. Our LOOCV analysis suggests that the model's predictions were worse for clinics where the CIP-R was higher than the national average. Second, implicit in our analysis is that equal weighting is given to ensuring treatment efficacy while also preventing the unnecessary use of CRO/CFX. However, in practice, a greater priority might be given to ensuring that patients receive an antibiotic that matches the susceptibility profile of their infection, causes the minimal side-effects and is inexpensive. This trade-off could be investigated through cost-effective analyses accounting for the consequences of prescribing ineffective treatment and any benefits gained from extending the clinical usefulness of existing antibiotics. Third, we included a limited number of features in our model (namely, the sexual behavior of the patient, geographic region, the prevalence of CIP resistance, and the change in regional prevalence of CIP resistance). However, other socio-demographic and health factors are believed to be associated with having a drug-resistant gonorrhea infection [13,19,22–28]. Including these variables may enhance our model's predictive power and could be considered in future analyses. Nevertheless, despite the limited range of features, our ML models displayed reasonable predictive capabilities and demonstrated the potential to reduce the unnecessary use of CRO/CFX. Moreover, the selected features can be easily obtained at the point of care, facilitating their implementation in clinical settings. However, we do acknowledge that knowing the features which are important to the models' prediction does not indicate how those features are being used by the model. The explainability of machine learning models is an area of ongoing and much-needed research to facilitate the use of machine learning models in practice [29,30]. Fourth, between 2000–2010, GISP included isolates only from the first 25 men at each surveillance site presenting with urethral gonorrhea. However, considering the potential variance in resistance dissemination among population groups, the performance of our ML models developed in this study may differ for other sexual behavior populations or individuals infected at anatomical sites beyond the urethra [13,19,31,32]. It is worth noting that GISP was expanded in 2017 to encompass isolates from multiple anatomical sites and from men and women

[18,33]. The incorporation of data from the enhanced Gonococcal Isolate Surveillance Project (eGISP) could mitigate this limitation in future analyses. Since 2020, the CDC has recommended single-drug therapy with ceftriaxone for the empiric treatment of gonorrhea [8]. As the prevalence of resistance to ceftriaxone is still extremely low (<0.002% among GISP isolates collected between 2000–2019), it would not be possible to develop models to predict the resistance to ceftriaxone given a patient's characteristics. However, the ML methods we described here could be used to guide the use of ceftriaxone in the empiric treatment of gonorrhea in the future when the resistance to this antibiotic starts to spread more widely. As suggested by our analysis, we anticipate that the treatment recommendation provided by ML models outperforms the ad-hoc and heuristic refinement of guidelines, as occurred during 2000–2007. Despite the global rise in AMR infections, the treatment of many bacterial infections (such as tuberculosis and gonorrhea) remains empiric due to a lack of access to drug-susceptibility testing (DST) or delays in receiving the results of DSTs. Our findings contribute to the emerging evidence that ML models trained on data from national AMR surveillance systems could identify resistance based on the patient's demographic and clinical information[34–40]. We showed that using predictive models to guide personalized treatment recommendations could reduce the use of newer antibiotics, and potentially extending their useful clinical lifespan. Future studies could investigate the potential of these ML models to inform the selection of antibiotics, at the point-of-care, for patients with AMR infections.

## Supporting information

**S1 File. S1: Supplementary methods.** S1.1 GISP data and data pre-processing. S1.2 Hyperparameter tuning. S1.3 Bootstrap algorithm to calculate confidence intervals for performance measures. S1.4 Model specificity and sensitivity. S1.5 Calculating the effectiveness of personalized treatment recommendations and their impact on the unnecessary use of CRO/CFX. **S2: Supplementary results.** S2.1 Permutation importance and feature selection. S2.2 Model performance across target years. S2.3 Percentage receiving effective treatment and unnecessary CRO/CFX for logistic regression and random forest. S2.4 Leave-one-out cross-validation for logistic regression and random forest
(DOCX)

## Acknowledgments

The authors would like to acknowledge CDC's Gonococcal Isolate Surveillance Project for the data used in these analyses. Additionally, the authors would like to thank the participating GISP sentinel sites and GISP regional laboratories for their contributions to GISP and collection of project data.

## Author Contributions

**Conceptualization:** Rachel E. Murray-Watson, Yonatan H. Grad, Reza Yaesoubi.

**Formal analysis:** Rachel E. Murray-Watson.

**Funding acquisition:** Reza Yaesoubi.

**Investigation:** Rachel E. Murray-Watson.

**Methodology:** Rachel E. Murray-Watson, Reza Yaesoubi.

**Project administration:** Reza Yaesoubi.

**Resources:** Reza Yaesoubi.

**Software:** Rachel E. Murray-Watson.

**Supervision:** Yonatan H. Grad, Reza Yaesoubi.

**Validation:** Rachel E. Murray-Watson, Reza Yaesoubi.

**Visualization:** Rachel E. Murray-Watson.

**Writing – original draft:** Rachel E. Murray-Watson, Yonatan H. Grad, Reza Yaesoubi.

**Writing – review & editing:** Rachel E. Murray-Watson, Yonatan H. Grad, Sancta B. St. Cyr, Reza Yaesoubi.

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
