## [Decision Letter · Decision Letter 0]

28 May 2024

PDIG-D-24-00036

Personalizing the empiric treatment of gonorrhea using machine learning models

PLOS Digital Health

Dear Dr. Murray-Watson,

Thank you for submitting your manuscript to PLOS Digital Health. After careful consideration, we feel that it has merit but does not fully meet PLOS Digital Health's publication criteria as it currently stands. Therefore, we invite you to submit a revised version of the manuscript that addresses the points raised during the review process.

Please submit your revised manuscript within 30 days Jun 27 2024 11:59PM. If you will need more time than this to complete your revisions, please reply to this message or contact the journal office at digitalhealth@plos.org. Please include the following items when submitting your revised manuscript:

We look forward to receiving your revised manuscript.

Kind regards,

Dhiya Al-Jumeily OBE, PhD

Section Editor

PLOS Digital Health

Journal Requirements:

1. We ask that a manuscript source file is provided at Revision. Please upload your manuscript file as a .doc, .docx, .rtf or .tex.

Additional Editor Comments (if provided):

Reviewers' comments:

Reviewer's Responses to Questions

**Comments to the Author**

1. Does this manuscript meet PLOS Digital Health’s publication criteria? Is the manuscript technically sound, and do the data support the conclusions? The manuscript must describe methodologically and ethically rigorous research with conclusions that are appropriately drawn based on the data presented.

Reviewer #1: Yes

Reviewer #2: Yes

2. Has the statistical analysis been performed appropriately and rigorously?

Reviewer #1: Yes

Reviewer #2: Yes

3. Have the authors made all data underlying the findings in their manuscript fully available (please refer to the Data Availability Statement at the start of the manuscript PDF file)?

Reviewer #1: Yes

Reviewer #2: No

4. Is the manuscript presented in an intelligible fashion and written in standard English?

Reviewer #1: Yes

Reviewer #2: Yes

5. Review Comments to the Author

Reviewer #1: Manuscript Number: PDIG-D-24-00036

Title: Personalizing the empiric treatment of gonorrhea using machine learning models

Abstract

- A clear overview of study objectives, methods, results, and conclusions.

- The significance of personalized treatment approaches is effectively highlighted.

- Specific data points were provided, but numerical values for improvements with personalized treatments compared to guidelines would enhance clarity.

Introduction

- Thorough contextualization of the research problem within the broader issue of AMR gonorrhea.

- A comprehensive overview of empirical treatment practices and the limitations of standardized guidelines.

- The historical context of treatment recommendations adds depth, but mentioning the limitations of current practices could improve discussion.

Methods

- Detailed and well-organized explanation of the dataset, study settings, predictors, outcomes, and model development.

- Justification for feature selection was provided, but the inclusion of handling missing data and potential biases would be beneficial.

- A brief rationale for choosing the neural network model as the primary focus would enhance clarity.

Results

- Effective presentation of key findings, including the prevalence of CIP resistance, model performance metrics, and impact of personalized treatment.

- Used figures and tables enhanced readability, and more detailed explanations of model performance metrics would aid interpretation.

Reviewer #2: The abstract should be rewritten to show the findings of the study – the present abstract only communicates background and procedural information with minimal results.

Some characteristics such as age group and ethnicity are important to include. Why have they not been considered.

Add list of definition to the method

What type of neural networks were used?

Page 5, there should not be a come after Python.

Justify the use of 10 fold cross validation

Page 7, on what basis was the target value chosen?

One of the limitations that should be highlighted is explainabilty of the models

---

## [Editor Report · Decision Letter 1]

11 Jun 2024

Personalizing the empiric treatment of gonorrhea using machine learning models

PDIG-D-24-00036R1

Dear Dr Murray-Watson,

We are pleased to inform you that your manuscript 'Personalizing the empiric treatment of gonorrhea using machine learning models' has been provisionally accepted for publication in PLOS Digital Health.

Best regards,

Dhiya Al-Jumeily OBE, PhD

Section Editor

PLOS Digital Health